# Necroptosis in pancreatic cancer promotes cancer cell migration and invasion by release of CXCL5

Yohei Ando[1], Kenoki Ohuchida[1]*, Yoshiki Otsubo[1], Shin Kibe[1], Shin Takesue[1], Toshiya Abe[1], Chika Iwamoto[2], Koji Shindo[1], Taiki Moriyama[3], Kohei Nakata[1], Yoshihiro Miyasaka[1], Takao Ohtsuka[1], Yoshinao Oda[4], Masafumi Nakamura[1]*

**1** Department of Surgery and Oncology, Graduate School of Medical Sciences, Kyushu University, Fukuoka, Japan, **2** Department of Advanced Medical Initiatives, Graduate School of Medical Sciences, Kyushu University, Fukuoka, Japan, **3** Department of Endoscopic Diagnostics and Therapeutics, Graduate School of Medical Sciences, Kyushu University, Fukuoka, Japan, **4** Department of Anatomical Pathology, Graduate School of Medical Sciences, Kyushu University, Fukuoka, Japan

\* kenoki@surg1.med.kyushu-u.ac.jp (KO); mnaka@surg1.med.kyushu-u.ac.jp (MN)

**Data Availability Statement:** All raw data files are available from the figshare database (URL; https:// figshare.com/articles/Raw_data_for_revised_ver_/

## Abstract

### Background

Necroptosis is a form of programmed cell death that is accompanied by release of intracellular contents, and reportedly contributes to various diseases. Here, we investigate the significance of necroptosis in pancreatic cancer.

### Methods

We used immunohistochemistry and western blot analysis to evaluate expression of the key mediators of necroptosis—receptor-interacting serine/threonine protein kinase 3 (RIP3) and mixed lineage kinase domain-like (MLKL)—in human pancreatic cancer. We also tested the effects of conditioned media (CM) from necroptotic cells on pancreatic cancer cells in Transwell migration and Matrigel invasion assays. Protein array analysis was used to investigate possible mediators derived from necroptotic cells.

### Results

RIP3 and MLKL are highly expressed in human pancreatic cancer tissues compared with normal pancreas. MLKL expression was particularly intense at the tumor invasion front. CM derived from necroptotic cells promoted cancer cell migration and invasion, but not CM derived from apoptotic cells. C-X-C motif chemokine 5 (CXCL5) was upregulated in CM derived from necroptotic cells compared with CM derived from control or apoptotic cells. Moreover, expression of the receptor for CXCL5, C-X-C-motif chemokine receptor-2 (CXCR2), was upregulated in pancreatic cancer cells. Inhibition of CXCR2 suppressed cancer cell migratory and invasive behavior enhanced by necroptosis.

11462274), DOI: https://doi.org/10.6084/m9.
figshare.11462274.v1

**Funding:** This work was supported in part by
Japan Society for the Promotion of Science Grants
in-Aid for Scientific Research (Grant Numbers
17H04284, 17K19602, 18H02881, 18K08708,
16H05418, and 17K19605). https://www.jsps.go.
jp/english/index.html The funder had no role in the
study design, data collection and analysis, decision
to publish, or preparation of the manuscript.

**Competing interests:** The authors have declared
that no competing interests exist.

## Conclusion

These findings indicate that necroptosis at the pancreatic cancer invasion front can promote
cancer cell migration and invasion via the CXCL5–CXCR2 axis.

## Introduction

Pancreatic cancer (PC) is one of the most lethal cancers in the world. It is the fourth leading
cause of cancer death, with a 5-year survival rate of less than 8% [1]. Although the best option
to improve prognosis is curative resection, PC is commonly diagnosed at a late stage [2]. Sev-
eral therapeutic options have been developed for metastatic PC, but satisfactory results have
yet to be obtained [3,4]. Clarification of mechanisms that promote PC malignancy, and devel-
opment of effective treatment strategies, are urgently needed.

Although cell death resistance is an important characteristic of cancer, various stresses can
cause cell death in tumors, such as hypoxia, nutrient deficiency, immune response and chemo-
therapy [5]. Cell death processes are largely of two types: programmed and non-programmed
cell death. Conventionally, apoptosis is considered to be programmed cell death, and necrosis
to be non-programmed [6]. These types of cell death have different morphological features.
Apoptosis is characterized by cytoplasmic shrinkage, nuclear condensation and the retention
of membrane and organelle integrity [7]. Necrosis is characterized by rapid cytoplasmic swell-
ing, rupture of the plasma membrane and release of intracellular contents [6,8]. Recently, how-
ever, parts of necrotic cell death were reported to also be regulated by defined molecular
pathways [9]. Regulated necrosis depends on the formation of the necrosome—termed
"necroptosis"—which requires receptor-interacting serine/threonine protein kinase (RIPK)-1/
3, and mixed lineage kinase domain-like (MLKL) [10–13]. Necroptosis can be triggered by
ligation of various death receptors, such as TNF receptor, TNF-related apoptosis-inducing
ligand (TRAIL) receptor, FAS (CD95) and Toll-like receptors [6,11]. In necroptosis signaling,
RIP1 form the cytosolic death-inducing signaling complex (DISC), which contains Fas-associ-
ated death domain (FADD), RIP3, and caspase-8 [14,15]. When caspase-8 is activated in
DISC, it can trigger apoptosis; when caspase-8 activity is then inhibited, RIP1 interacts with
RIP3, becomes phosphorylated, and assembles the necrosome [15,16]. Phosphorylated RIP3
induces MLKL phosphorylation and formation of oligomers [17]. Oligomerized MLKL trans-
locates to the plasma membrane and forms membrane-disrupting pores [6,17,18]. As with
necrosis, necroptotic cells release intracellular contents that affect the surrounding cells and
environment [19,20]. Recent studies suggest that necroptosis plays both positive and negative
roles in development and progression of cancer, but its role in PC is unclear [21]. Pancreatic
cancer is associated with potential inducers of necroptosis, such as TNFR1, TRAIL receptors
and FAS [22–25]. Here, we investigated the significance of necroptosis in PC.

In this study, we have shown that PC has potential to induce necroptosis, and CXCL5
released from necroptotic cells promotes cancer cell migration and invasion via CXCR2 in PC
cells.

## Materials and methods

### Human pancreatic tissue samples

Pancreatic cancer tissues were obtained from 21 patients who had undergone resections at
Kyushu University Hospital. The study was approved by the Ethics Committee of Kyushu

University and conducted according to the Ethical Guidelines for Human Genome/Gene Research enacted by the Japanese Government and the Helsinki Declaration.

## Immunohistochemistry and evaluation

Immunohistochemical staining was performed as described [26]. Formalin-fixed, paraffin-embedded tissue was cut at 4-μm thicknesses and deparaffinized with xylene and ethanol. Endogenous peroxidase activity was blocked by methanol containing 0.3% hydrogen peroxidase. Antigen retrieval was performed by boiling in a microwave oven (citrate buffer, pH 6.0). The sections were incubated overnight at 4˚C with primary antibodies against anti-RIP3 (ab56164; Abcam, Cambridge, UK; 1:100), anti-MLKL (ab194699; 1:100), anti-CXCR2 (ab65968; 1:200), anti-CXCL5 (ab9802; 1:500) and anti-CXCL1 (ab86436; 1:100). The immune complexes were then visualized using Envision Detection System (Dako, California, USA) and 3,3′-diaminobenzidine (DAB) Kit (Dako). DAB intensity of MLKL was evaluated as previously described using ImageJ software [27]. Images were adjusted by subtracting background as RGB values close to 255 in empty area. Tumor area was selected and mean intensity of DAB color spectrum was calculated, ranging 0 (black) to 255 (total white). The final DAB intensity was calculated according to the formula $f = 255 - i$, where $f$ = final DAB intensity, $i$ = mean DAB intensity. Intensity was calculated in 5 fields at the borders and cores from each patient at 200× magnification under a light microscope.

## Cell lines and culture conditions

The human pancreatic cancer cell lines AsPC-1 (American Type Culture Collection [ATCC], Manassas, VA, USA), BxPC-3 (ATCC), Capan-1 (ATCC), KP-2 (National Institutes of Biomedical Innovation, Health and Nutrition [NIBIOHN], Osaka, Japan), MIA PaCa-2 (NIBIOHN), PANC-1 (RIKEN BioResource Center, Ibaraki, Japan) and SUIT-2 (NIBIOHN) were purchased. Human pancreatic duct epithelial cells (HPDE) was the gift of the Dr M.-S. Tsao, University of Toronto, Canada. HPDE was maintained in HuMedia-KG2 (KURABO, Osaka, Japan) at 37˚C in a humidified atmosphere containing 10% $CO_2$. Other cells were maintained in DMEM (Sigma Chemical Co., St. Louis, MO, USA) supplemented with 10% fetal bovine serum (FBS), streptomycin (100 μg/ml), and penicillin (100 units/ml) at 37˚C with humidified 90% air and 10% $CO_2$ [28].

## Western blot analysis

Anti-RIP3 (ab56164; 1:800), anti-MLKL (ab194699; 1:1000), anti-MLKL (phospho S358) (ab187091; 1:1000), anti-caspase-8 (NB100-56116; Novus Biologicals, Colorado, USA 1:1000), anti-CXCR2 (ab65968; 1:500), anti-CXCL5 (ab9802; 1:1000) and anti-β-actin (ab8227; 1:2000) antibodies and then probed with secondary antibodies conjugated to horseradish peroxidase (Santa Cruz Biotechnology, Santa Cruz, CA, USA). Immunoblots were detected by enhanced chemiluminescence with ChemiDocXRS (Bio-Rad Laboratories). Each experiment was repeated more than three times.

## Cell death induction and collection of conditioned media

To induce necroptosis, cells were treated with a combination of recombinant human tumor necrosis factor-α (TNF-α) (Peprotech, New Jersey, USA; 10 ng/ml), second mitochondrial-derived activator of caspases (SMAC) mimetic (BV6; Selleck Chemicals, Houston, USA; 1 nM) and pan-caspase inhibitor (zVAD-FMK; ENZO Life Science, New York, USA; 40 μM). To induce necroptosis in AsPC-1 cells with another combined necroptotic stimulus, we used

recombinant human Fas ligand (Peprotech; 50 ng/ml), SMAC mimetic (LCL161; Selleck Chemicals; 2 μM) and caspase-8 inhibitor (zIETD-FMK; Enzo; 2 μM). To induce apoptosis, cells were treated with TNF-α (10 ng/ml) and BV6 (1 nM). To inhibit necroptosis, necrostatin-1 (Enzo; 30 μM) was added an hour before treating with the above agents.

To evaluate the effects of chemotherapeutic agents on chemokine receptor expression in PC cells, we treated PC cells by paclitaxel (FUJIFILM Wako Chemicals, Osaka, Japan; 40nM) or paclitaxel (40nM) + necrostatin-1 (30μM) for 12hours.

To collect CM of cells in which necroptosis or apoptosis were induced, after treatment for 3 hours with the aforementioned agents, cells were washed twice with PBS and their media replaced with fresh media. After 12 hours' incubation at 37˚C, CM was collected and filtered with a 22-μm syringe filter (Merck, Darmstadt, Germany). After being centrifuged at 1500 rpm for 5 minutes, supernatants were collected and stored 4˚C until used.

## Cell death analysis

To evaluate the percentage of dead cells, 100 μl at $1\times10^5$ cells/ml were seeded on 96-well plates and treated with stimulants for 15 hours. After treatment, EthD-Ⅲ (PromoKine, Heidelberg, Germany; 1.6 μM) and Hoechst 33342 (Sigma-Aldrich Co.; 2 μM) were added shortly before image acquisition. Fluorescent images were acquired using a fluorescence microscope (BZ-9000; Keyence, Osaka, Japan). After 15 hours' treatment with stimulants, dead cells were positive for EthD-Ⅲ, and all cells were positive for Hoechst 33342. All images were analyzed in ImageJ. Each experiment was repeated at least three times with five wells per condition.

For morphological analysis, cells were treated with stimulants for 6 hours, and stained by EthD-Ⅲ, Annexin V-FITC (MBL, Nagoya, Japan) and Hoechst 33342. Living cells have normal-shaped nuclei and are negative for EthD-Ⅲ. Apoptotic cells have strongly condensed or fragmented nuclei and are positive for Annexin V, but negative for EthD-Ⅲ. Necrotic or necroptotic cells have normal-shaped nucleus and are positive for both Annexin V and EthD-Ⅲ. Late apoptotic cells are positive for EthD-Ⅲ, but have condensed and fragmented nuclei.

## Wound healing assay

Wound healing assays were performed using culture inserts (80209, Ibidi, Martinsried, Germany), as previously described [29,30]. Briefly, cells were seeded at 70 μl/chamber at a density of $5\times10^5$ cells/ml. Cells were cultured at 37˚C overnight to form a confluent monolayer prior to careful removal of the insert. After control CM or CM-TSZ from PC cells was added, time-lapse imaging was performed for 6 hours using a fluorescent microscope (BZ-9000; Keyence). Each experiment was repeated more than three times.

## Transwell migration assay, Matrigel invasion assay

The Transwell migration and Matrigel invasion assays were performed as previously described [31,32]. For migration assays, uncoated Transwell inserts with 8-μm pores (#353097, BD Falcon, Franklin Lakes, NJ, USA) were inserted into 24-well plates. We added 500 μl of DMEM with 10% FBS to each lower chamber, and placed $4\times10^4$ BxPC-3 or PANC-1 cells, or $1\times10^5$ AsPC-1 cells, resuspended in 250 μl of the same medium in each upper chamber with conditioned media (CM) derived from PC cells or recombinant human CXCL5 (Peprotech; 10 ng/ml). After 24 hours' incubation at 37˚C, migrated cells were counted in five random fields at 200× magnification using ImageJ software. The results were expressed as the mean number of migrated cells per field.

For the Transwell-Matrigel invasion assays, the Transwell inserts were coated with 20 μg/well Matrigel (356234, Corning, Maine, USA). We suspended $1\times10^5$ AsPC-1 or BxPC-3 cells,

or $4\times10^4$ PANC-1 cells, in 250 μl of medium, with CM from PC cells or rh CXCL5, and placed them in the upper chamber. Plates were incubated for 36 hours for BxPC-3 cells, and 48 hours for AsPC-1 or PANC-1 cells. To inhibit CXCR2, we pretreated cells with SB225002 (Selleck; 10 nM) for one hour. Each experiment was repeated more than three times.

### 3D spheroid invasion assay

The 3D spheroid invasion assay was a modification of a previously described method [33]. To form tumor spheroids, $2.5\times10^3$ cells were cultured in ultra-low attachment 96-well plates with round bottoms (Corning) for 4 days. The spheroids were then treated with TSZ ± Inhibitor of RIPK1 kinase necrostatin-1 (nec-1), or with DMSO (control) for 3 hours. After treatment, medium was replaced with 50 μl of type I collagen gel (final concentration: 2 mg/ml, Cat no. 354236; BD Biosciences). An hour later, 100 μl of medium was gently added and incubated at 37˚C. Images were acquired using fluorescence microscope (BZ-9000; Keyence) at 0, 6, 12, 24 and 48 hours. To evaluate the invasion area, the newly invaded area was compared with the area at 0 hours. Image analysis was performed using ImageJ software. For cell-death analysis, EthD-III (final concentration; 1.6 μM) was added to each well an hour before image acquisition. Each experiment was repeated more than three times.

### Proliferation assay

Cell proliferation was evaluated using a cell counting Kit-8 (CCK-8, Dojindo, Kumamoto, Japan). Cells were seeded in 96-well plates at a density of $8\times10^3$/well. After the cells attached to the plates, CM from PC cells or SB225002 (10 nM) was added and incubated. Three hours after adding CCK-8, the optical density at 450-nm absorbance was detected by microplate reader (TECAN, Kanagawa, Japan). Each experiment was repeated more than three times.

### Enzyme-linked immunosorbent assay (ELISA)

AsPC-1 and BxPC-3 cells were treated with TSZ ± nec-1 or with DMSO (control) for 3 hours. Cells were then washed twice with PBS and placed in serum-free medium. After 12 hours' incubation, CM was collected, and CXCL5 was quantified using a Human CXCL5 Quantikine ELISA Kit (R & D Systems Europe Ltd, Oxford, UK).

### Real-time quantitative reverse transcription polymerase chain reaction (qRT-PCR)

We performed qRT-PCR using the iTaq Universal SYBR-Green One-Step kit and CFX96 Touch Real-Time PCR Detection systems (Bio-Rad Laboratories). Primers were purchased from Takara Bio (Kusatsu, Japan). Human 18S ribosomal RNA was used as the endogenous control gene. The following primers were used in the present study: C-X-C motif chemokine receptor 2 (*CXCR2*), forward, 5′-TCTTCAGGGCACACTTCCACTAC-3′ and reverse, 5′-GGG CTGCATTGACACTGAGA-3′; C-C motif chemokine receptor 6 (*CCR6*), forward, 5′- CAGT CAACAAGCCTGACCCTGTA-3′ and reverse, 5′-AACACCTGTTCTGCCATTGTCC-3′; 18S, forward, 5′-ACTCAACACGGGAAACCTCA-3′ and reverse, 5′-AACCAGACAAATCGCTCC AC-3′. Each experiment was repeated more than three times.

### Silencing of CXCR2 by small-interfering RNA

Two siRNAs that target CXCR2 (Cat no. SI03093979, SI00447097; Qiagen, Venlo, Netherlands) and non-targeting siRNA (Cat no. 1027310; Qiagen) were purchased. Transfection was performed by electroporation using a Nucleofector System (Lonza, Basel, Switzerland).

according to the manufacturer's recommendations. Transfected cells were used in subsequent experiments 48–72 hours after transfection.

## Protein array analysis

We purchased a human cytokine antibody array (120 targets; ab193656). To prepare necroptosis CM, AsPC-1 cells were treated with TSZ or DMSO (control) for 3 hours, then washed twice with PBS and placed in serum-free medium. After 12 hours' incubation, CM was collected. Images were acquired using a ChemiDoc XRS (Bio-Rad Laboratories) and analyzed using ImageJ software protein array analyzer.

## Statistical analysis

Data are shown as mean ± standard error (SE). Comparisons between two groups were performed by Student's $t$ tests, $P < 0.05$ was considered significant. All statistical analyses were carried out using JMP Pro 11 software (SAS Institute, Cary, NC, USA).

## Results

### Key mediators of necroptosis were expressed in human PC

To examine whether the necroptosis can occur in human PC, we performed immunohistochemistry tests for key mediators of necroptosis signaling in human PC tissues. Patients' characteristics are shown in Table 1. Expression of RIP3 and MLKL were significantly greater in human PC tissue than in surrounding normal pancreatic tissue (Fig 1A). Interestingly, we found that MLKL intensity was higher in the invasive front of tumor than in the center (Fig 1B and 1C). Western blotting confirmed that MLKL expression was greater in human PC cells than in HPDE (Fig 1D).

### TNF-α, SMAC mimetic and zVAD-FMK treatment induced necroptosis in PC

We induced necroptosis in PC cells using a combination of human recombinant TNF-α, SMAC mimetic, and zVAD-FMK (TSZ) [34,35]. TSZ treatment induced cell death in PC cells with RIP3 and high MLKL expression, such as AsPC-1, BxPC-3 and Capan-1 cells. Furthermore, the specific inhibitor of necroptosis, necrostatin-1 (nec-1), prevented TSZ treatment-induced cell death (Fig 2A and 2B). However, PC cells with low MLKL expression, such as PANC-1, KP-2 and SUIT-2 cells, or cells with low RIP3 expression, such as MIA PaCa-2, were insensitive to necroptotic stimuli. Instead, TNF-α and SMAC mimetic (TS) treatment induced cell death in PANC-1, KP-2 and MIA PaCa-2 cells (Fig 2A and 2B). SUIT-2 was insensitive to any stimulant.

To evaluate the morphological features of dead cells, we performed combination staining with Hoechst 33342, EthD-III and Annexin V. Cells that underwent necroptosis had normally shaped nuclei, and were positive for EthD-III and Annexin V. BxPC-3 cells treated with TSZ for 6 hours showed features of necroptosis. Cells in early-stage apoptosis are positive for Annexin V, but negative for EthD-III. PANC-1 cells treated with TS for 6 hours showed features of early-stage apoptosis (Fig 2C). The direct executor of necroptosis, p-MLKL, was upregulated in TSZ treated BxPC-3 and AsPC-1 cells. The activator of the apoptosis signal pathway, cleaved caspase-8 p43/41, was upregulated in TS-treated PANC-1 cells (Fig 2D). These findings suggest that human PC cells with RIP3 and high MLKL expression have high potential for inducing necroptosis. PC cells with RIP3 and low MLKL expression have potential for inducing apoptosis.

**Table 1. Clinicopathological characteristics for the 21 patients of this study.**

| Patient | age | sex | pT stage | pN stage | Histologic grade | UICC stage | neoadjuvant therapy |
|---------|-----|--------|----------|----------|------------------|------------|---------------------|
| 1 | 55 | Female | 3 | 1 | G2 | IIB | none |
| 2 | 63 | Female | 3 | 1 | G3 | IIB | none |
| 3 | 85 | Female | 3 | 1 | G3 | IIB | none |
| 4 | 78 | Female | 3 | 0 | G2 | IIA | none |
| 5 | 50 | Female | 3 | 1 | G2 | IIB | none |
| 6 | 55 | Female | 3 | 1 | G3 | IIB | none |
| 7 | 72 | Male | 3 | 0 | G2 | IIA | none |
| 8 | 80 | Male | 3 | 1 | G2 | IIB | none |
| 9 | 73 | Female | 4 | 1 | G2 | III | none |
| 10 | 62 | Male | 3 | 1 | G3 | IIB | none |
| 11 | 70 | Male | 3 | 1 | G3 | IIB | none |
| 12 | 46 | Male | 3 | 1 | G3 | IIB | none |
| 13 | 68 | Female | 3 | 1 | G3 | IIB | none |
| 14 | 73 | Male | 3 | 1 | G3 | IIB | none |
| 15 | 57 | Male | 3 | 0 | G3 | IIA | none |
| 16 | 59 | Male | 3 | 1 | GX | IIB | none |
| 17 | 65 | Male | 3 | 1 | G3 | IIB | GnP |
| 18 | 65 | Male | 3 | 1 | G2 | IIB | GnP |
| 19 | 34 | Female | 3 | 1 | G3 | IIB | GnP |
| 20 | 63 | Female | 3 | 1 | G3 | IIB | GnP |
| 21 | 43 | Female | 3 | 1 | G3 | IIB | GnP |

Tumor classification and stage refer to the 7th edition of UICC on cancer staging system. UICC, Union for International Cancer Control.

GnP; gemcitabine + nab-paclitaxel

## Conditioned medium from necroptotic cells promoted PC cell migration and invasion

To investigate the effects of necroptosis on migration of PC cells, we induced necroptosis in the PC cells with TSZ and then used the resulting conditioned media (CM) in wound healing and Transwell migration assays. In both assays, CM from TSZ-treated (CM-TSZ) BxPC-3 and AsPC-1 cells showed significantly greater motility (Fig 3A–3D). In the Transwell-Matrigel invasion assay, CM-TSZ from BxPC-3 and AsPC-1 cells significantly increased the number of invading cancer cells (Fig 3E and 3F). Interestingly, CM from TS-treated PANC-1 cells, in which apoptosis alone was induced without necroptosis, did not enhance cancer cell migration and invasion (Fig 3B, 3D and 3F). Neither CM-TSZ nor CM-TS affected cancer cell proliferation ability (Fig 3G). Next, to determine the effect of necroptosis on the local microenvironment, we established a 3D spheroid model of PC and induced necroptosis by incubating them with TSZ for three hours. EthD-III staining revealed that outer side of spheroid cells were in a state of induced death (Fig 3H). Spheroids in which necroptosis was induced showed highly invasive behavior compared with controls (Fig 3I and 3J). These findings indicate that necroptosis in PC promotes migration and invasion.

## Macrophage inflammatory protein-3 alpha (MIP-3α) and C-X-C motif chemokine 5 (CXCL5) expression were specifically upregulated in CM from necroptotic cells

Necroptosis is accompanied by the release of immunogenic intracellular contents. To investigate possible necroptotic cell-derived mediators, which promote PC cell migration and

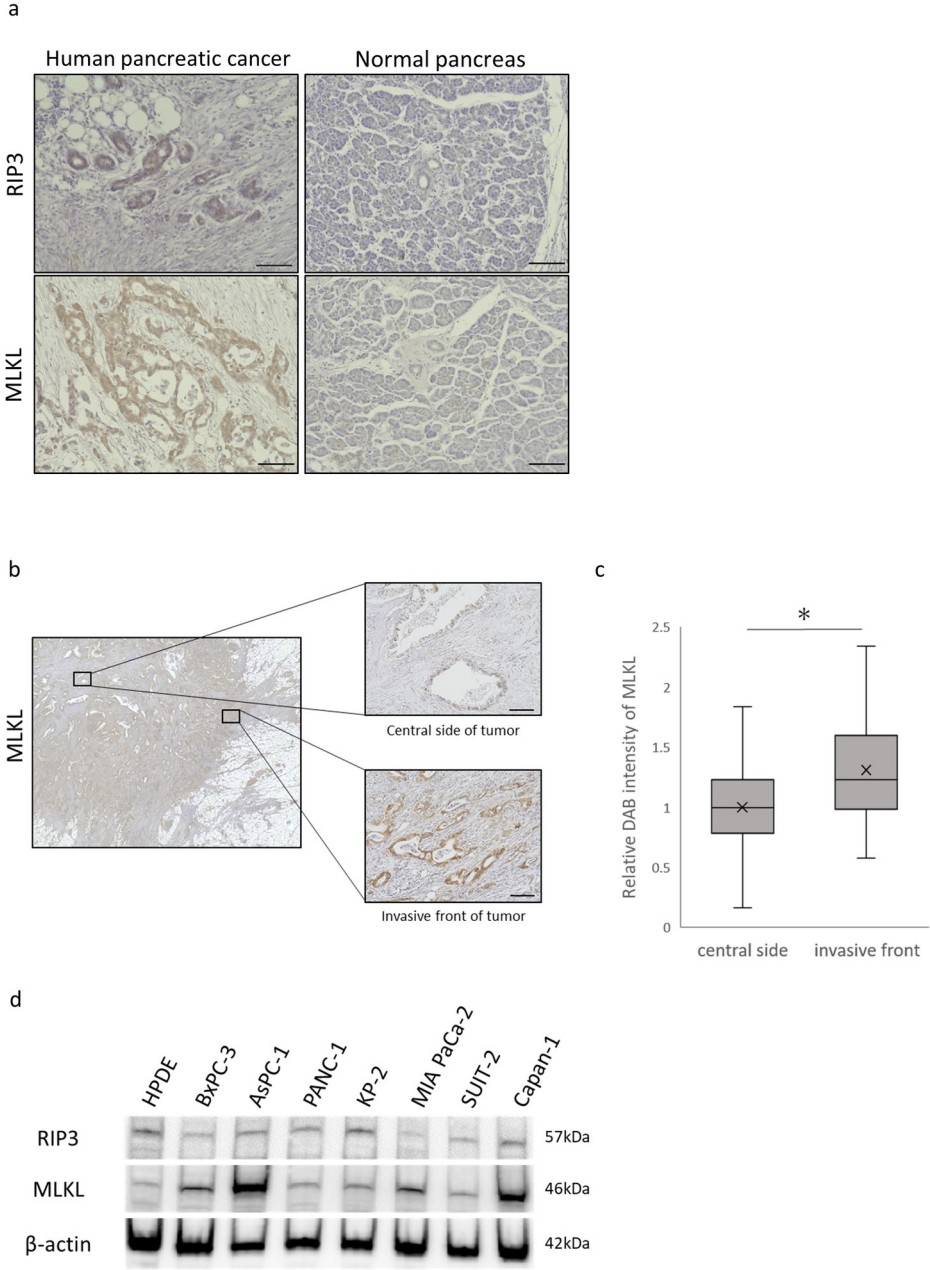

**Fig 1. Detection of RIP3 and MLKL, key mediators of necroptosis in human pancreatic cancer.** (a) RIP3 and MLKL immunohistochemistry in resected specimens of human pancreatic cancer and surrounding normal pancreatic tissues (scale bars = 100 μm). (b) Representative images of MLKL expression at the invasive front and the center of the tumor (scale bars = 100 μm). (c) DAB intensity of MLKL in pancreatic cancer cells was significantly higher at the tumor invasive front than at the center. Five fields at a magnification of 200× per 21 patients were analyzed. (d) Western blot analyses of RIP3 and MLKL in human pancreatic cancer cells and HPDE. *P<0.01.

invasion, we performed a protein array analysis of 120 cytokines. Compared with control CM, MIP-3α, CXCL5 and interleukin-8 (IL-8) were upregulated in CM-TSZ from AsPC-1 cells (Fig 4A and 4B). However, we also found IL-8 was upregulated in apoptotic cell-derived PANC-1 CM (Fig 4C and 4D). We then evaluated mRNA expression of the chemokine receptors whose ligands were upregulated in CM-TSZ. CXCR2, which is the receptor for CXCL5,

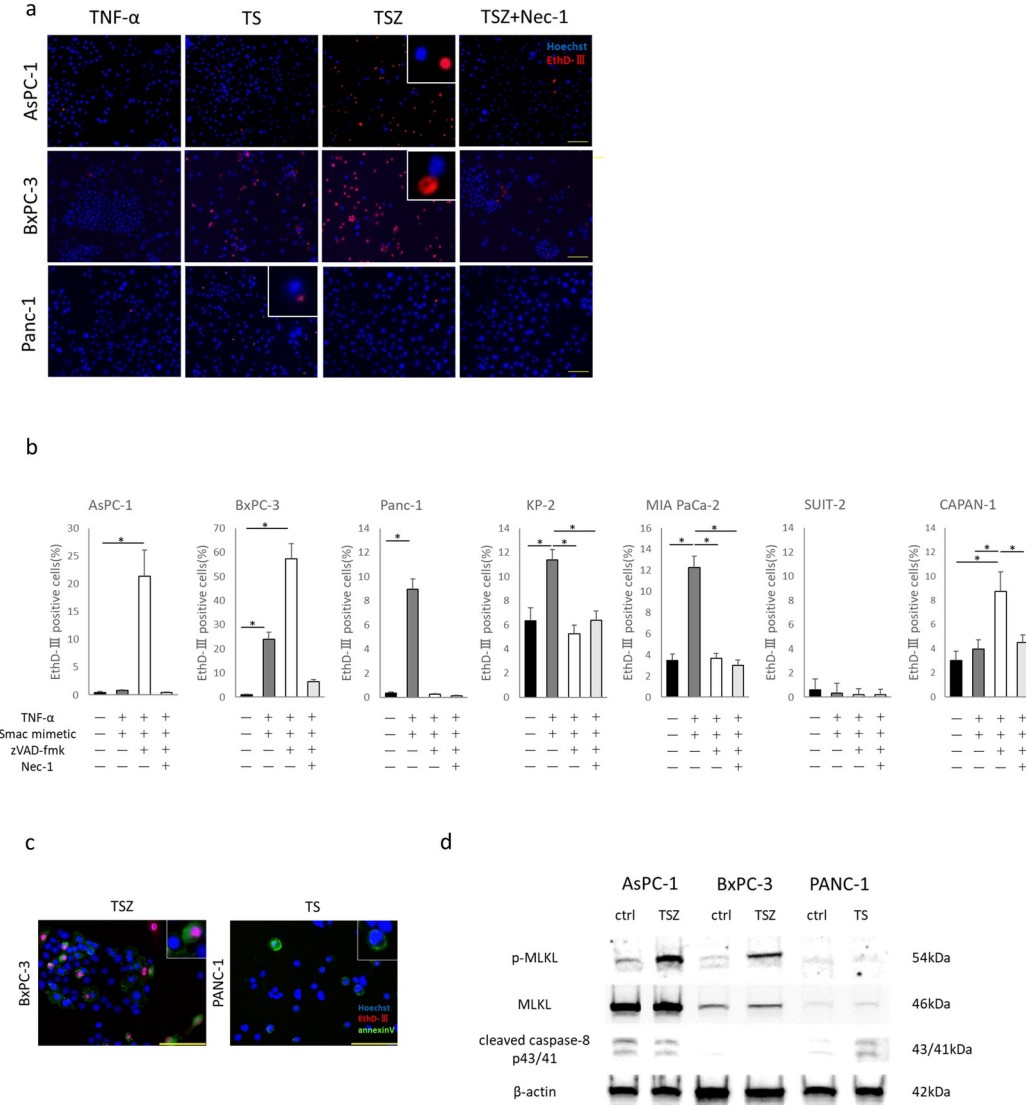

**Fig 2. Induction of necroptosis in pancreatic cancer cells.** (a) Fluorescent images of pancreatic cancer cells treated with TNF-α, Smac mimetic, zVAD-FMK and necrostatin-1 (nec-1). TS: TNF-α + smac mimetic; TSZ: TS + zVAD-FMK. Cells stained by Hoeschst 33342 and EthD-Ⅲ. EthD-Ⅲ staining (red) indicates dead cells (scale bars = 100 μm). (b) Rate of dead cells treated with each agent for 12 hours. (c) Morphological features of dead cells. Pancreatic cancer cells were treated with TSZ or TS for 6 hours. After treatment, cells were stained by Hoechst 33342 (blue), EthD-Ⅲ (red) and Annexin V (green; scale bars = 100 μm). (d) Western blot analysis of p-MLKL (direct activator of necroptosis), MLKL and cleaved caspase-8 p43/41 (activator of apoptosis signal pathway). Graph shows mean ± SE. *$P<0.01$.

was significantly overexpressed in PC cells compared with non-cancerous HPDE cells (Fig 4E). Expression of mRNA for the MIP-3α receptor of CCR6 did not significantly differ between AsPC-1 and HPDE cells (Fig 4F). Moreover, CCR6 expression was remarkably low in BxCP-3 cells (Fig 4F). Western blot analysis confirmed that CXCR2 expression was higher in PC cells than in HPDE (Fig 4G). Therefore, in the following experiments, we focused on CXCL5. ELISA showed that CXCL5 protein was upregulated in CM-TSZ derived from AsPC-1 and BxPC-3 cells, and that nec-1 prevented CXCL5 upregulation in CM-TSZ (Fig 4H). To confirm the upregulation of CXCL5 was caused by necroptosis, but not by agents, we tested another necroptotic stimulus. A combination of Fas ligand, LCL161 and zIETD-FMK (FLZ

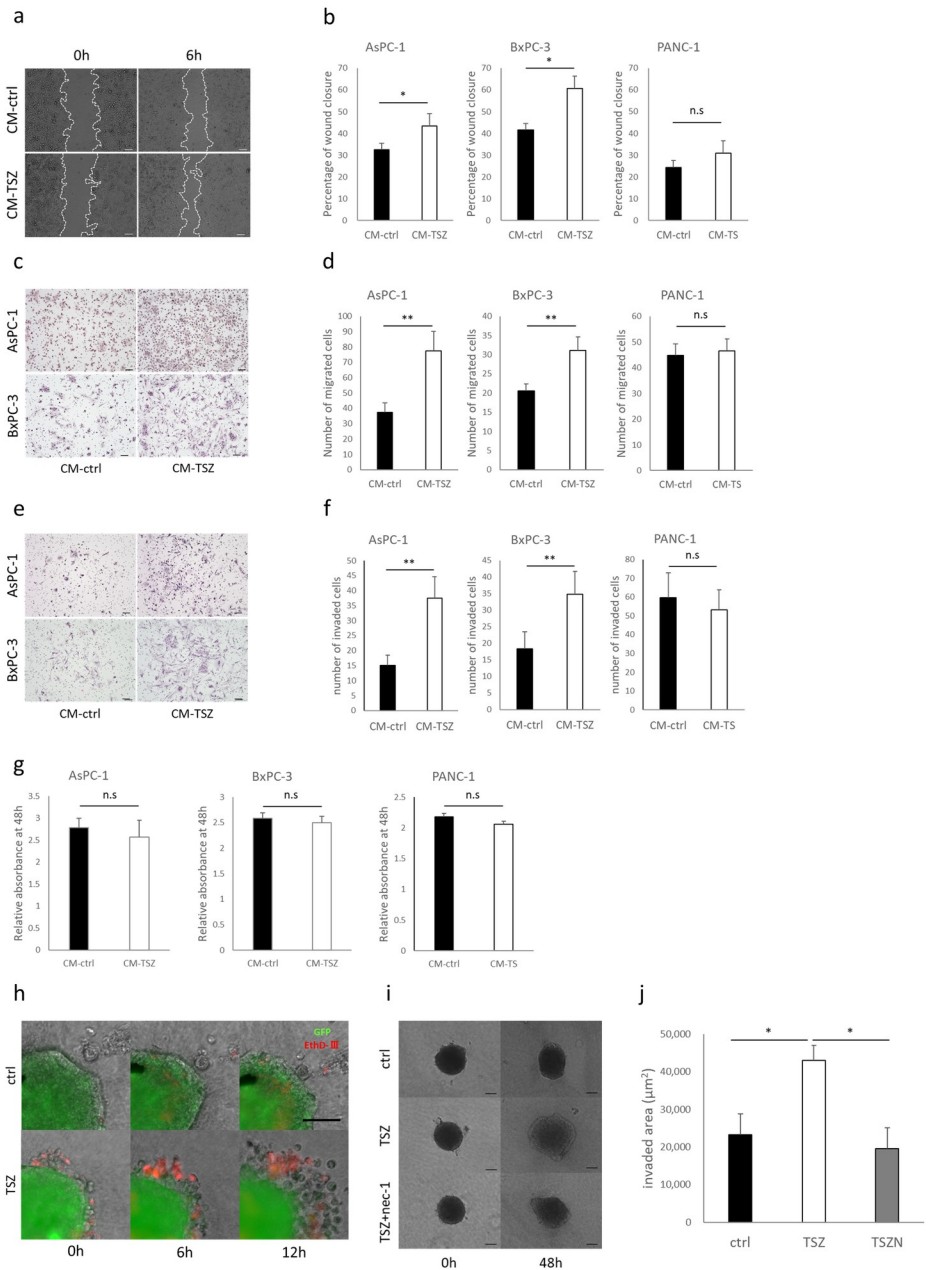

**Fig 3. Effects of conditioned medium derived dead cells on pancreatic cancer.** Wound healing assay; (a) representative images of AsPC-1 wound closure, (b) quantitative data (scale bars = 100 μm). Transwell migration assay; (c) representative images of migrated cells, (d) quantitative data of migrated cells (scale bars = 100 μm). Transwell-Matrigel invasion assay; (e) representative images of invaded cells, (f) quantitative data of invaded cells (scale bars = 100 μm). (g) CCK-8 proliferation assay; effect on pancreatic cancer cell proliferation by conditioned medium derived from dead cells or DMSO (control) after 48 hours. Absorbance relative to 0 hour. (h) Images of dead cell localization and subsequent cell invasion at 0, 6, and 12 hours. BxPC-3 spheroids (green) were treated with TSZ or DMSO (control) for 3 hours, and stained by EthD-Ⅲ (scale bars = 100 μm). (i) Representative images of 3D spheroid invasion assay. (j) Quantitative data for spheroid-invaded area. Graphs show mean ± SE. $^*P < 0.05$; $^{**}P < 0.01$; n.s; not significant.

treatment) induced necroptosis in AsPC-1 cells, as shown by increased cell death and upregulated p-MLKL (S1A–S1C Fig). CXCL5 upregulation in CM-FLZ from AsPC-1 was detected by

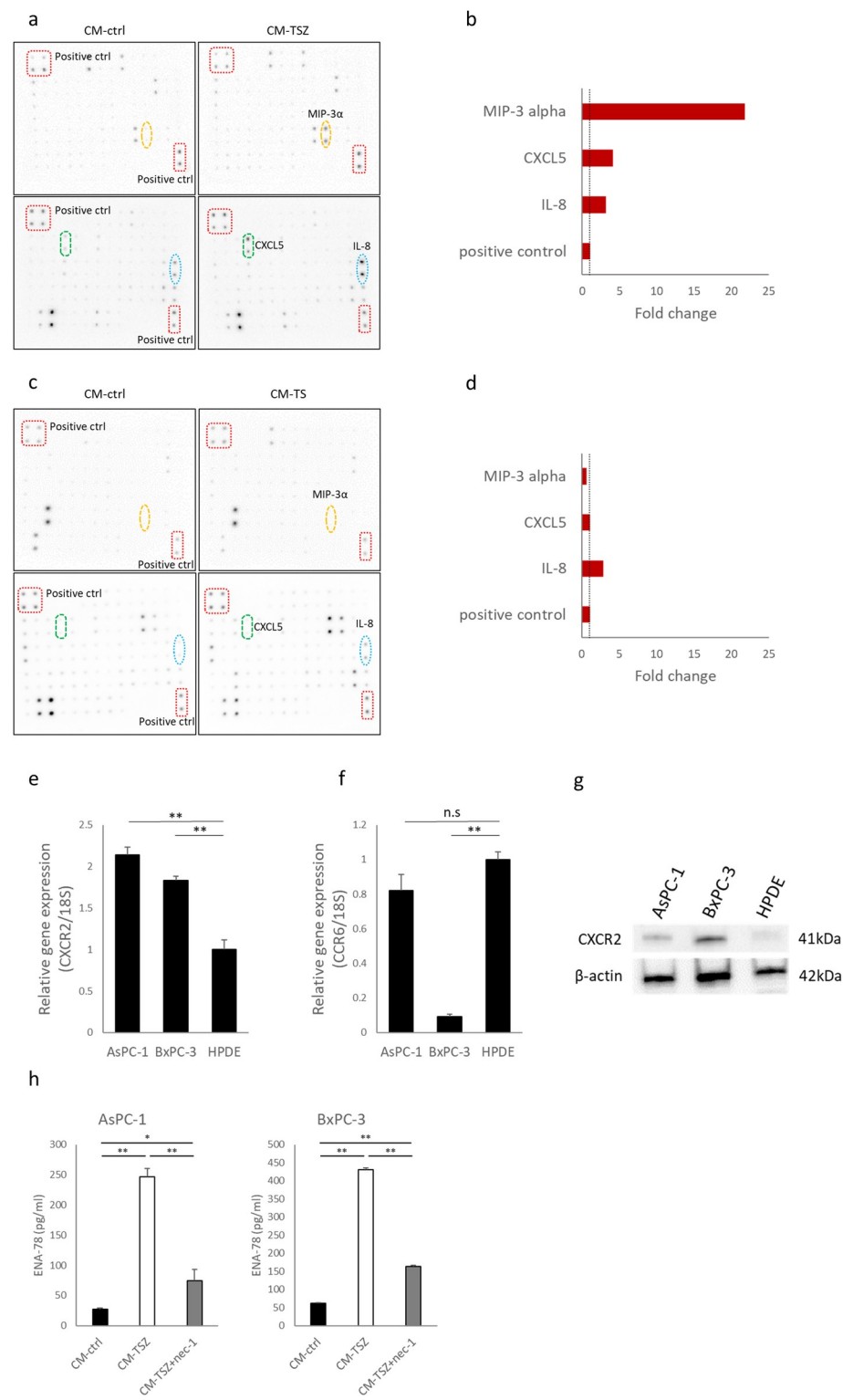

**Fig 4. Detection of cytokines in conditioned medium from necroptotic cells.** (a) Images of human cytokine antibody array (120 targets) of conditioned medium derived from AsPC-1 treated with necroptotic stimuli or DMSO (control). Boxes: positive controls; circles: CXCL5, MIP-3α and IL-8. (b) Signals were quantified relative to CM-control. (c) Human cytokine antibody array of conditioned medium from PANC-1, which was treated with apoptotic stimuli or DMSO (control). (d) Signals relative to CM-control. (e, f) Analysis of mRNA expression of chemokine receptors, *CXCR2* and *CCR6* by qRT-PCR. Results are shown relative to gene expression in non-cancerous HPDE cells

after normalization against 18S rRNA. (g) Western blot analysis of CXCR2 in human pancreatic cancer cells and in HPDE. (h) Concentration of CXCL5 in conditioned medium from AsPC-1 or BxPC-3, which were treated with TSZ ± nec-1 or DMSO (control), and measured by ELISA. Graphs show mean ± SE. *$P < 0.05$; **$P < 0.01$.

ELISA (S1D Fig). These findings indicate that necroptotic PC cells release CXCL5 and may have some effect on PC cells via the receptor CXCR2.

To clarify whether the CXCL5 released into the conditioned medium after death-inducing stimulus was stored in cells or induced in cells undergoing programmed cell death, we treated PC cells with TSZ and detected temporal changes of CXCL5 levels in PC cells. CXCL5 expression in PC cells was increased after TSZ treatment (S1E Fig). These results suggest that CXCL5 was induced in the process of programmed cell death and released into the conditioned medium.

Next, we observed expression patterns of CXCR2 and CXCL5 in human PC tissues. Consistent with a previous report[36], CXCR2 expression was evident at the invasive front of the tumor (S2A and S2B Fig). Although CXCL5 expression varied among patients (S2C Fig), we did not observe significant changes due to localization or preoperative chemotherapy. CXCL1 is a secreted growth factor that signals through CXCR2. We assessed CXCL1 expression in PC tissues. Interestingly, CXCL1 was mainly expressed in stromal cells around the tumor than in tumor cells (S2D Fig). Stromal cells expressing CXCL1 may have some effect on tumors via CXCR2. To clarify the influence of chemotherapy on the PC cell expression pattern of CXCR2 and MLKL we compared patients who received or did not receive neoadjuvant chemotherapy (NAC). CXCR2 expression at the invasive front was slightly reduced in patients treated with neoadjuvant chemotherapy, but changes of MLKL expression was not significant (S2E and S2F Fig).

## Chemokine released from necroptotic cells promoted cancer cell migration via C-X-C-motif chemokine receptor 2 (CXCR2)

Next, we examined the involvement of CXCR2 in the process of necroptosis that promotes cancer cell migration and invasion. In the transmembrane migration and Matrigel invasion assays, treatment with the selective CXCR2 antagonist, SB225002, impeded migratory and invasive behavior enhanced by necroptosis (Fig 5A–5C). SB225002 did not influence proliferation of BxPC-3 and AsPC-1 cells (Fig 5D). We performed RNA interference using two specific siRNAs for *CXCR2*. Western blot analysis confirmed knockdown of *CXCR2* (Fig 5E). Knockdown of *CXCR2* impeded migratory and invasive behavior enhanced by CM-TSZ in both AsPC-1 and BxPC-3 cells (Fig 5F–5J). Furthermore, recombinant human CXCL5 enhanced migratory and invasive behavior in AsPC-1 and BxPC-3 cells (Fig 6A–6C). These findings suggest that CXCL5, which is released from necroptotic PC cells, promotes cancer cell migration and invasion via CXCR2.

## Necroptosis-inducing chemotherapeutic agent changes CXCR2 expression in PC cells

Paclitaxel (PTX) has been reported to induce necroptosis in cancer [37]. We treated PC cells with PTX or PTX + nec-1 for 12hours and then detected changes in CXCR2 expression. PTX treatments induced cell death and p-MLKL expression in AsPC-1 and BxPC-3 cells, but inhibition by nec-1 was not significant (Fig 7A, 7B and 7D). We did not detect significant changes in CXCR2 mRNA expression levels after these treatments (Fig 7C). However, in western blot analysis CXCR2 expression was reduced in AsPC-1 cells by these treatments, but increased in

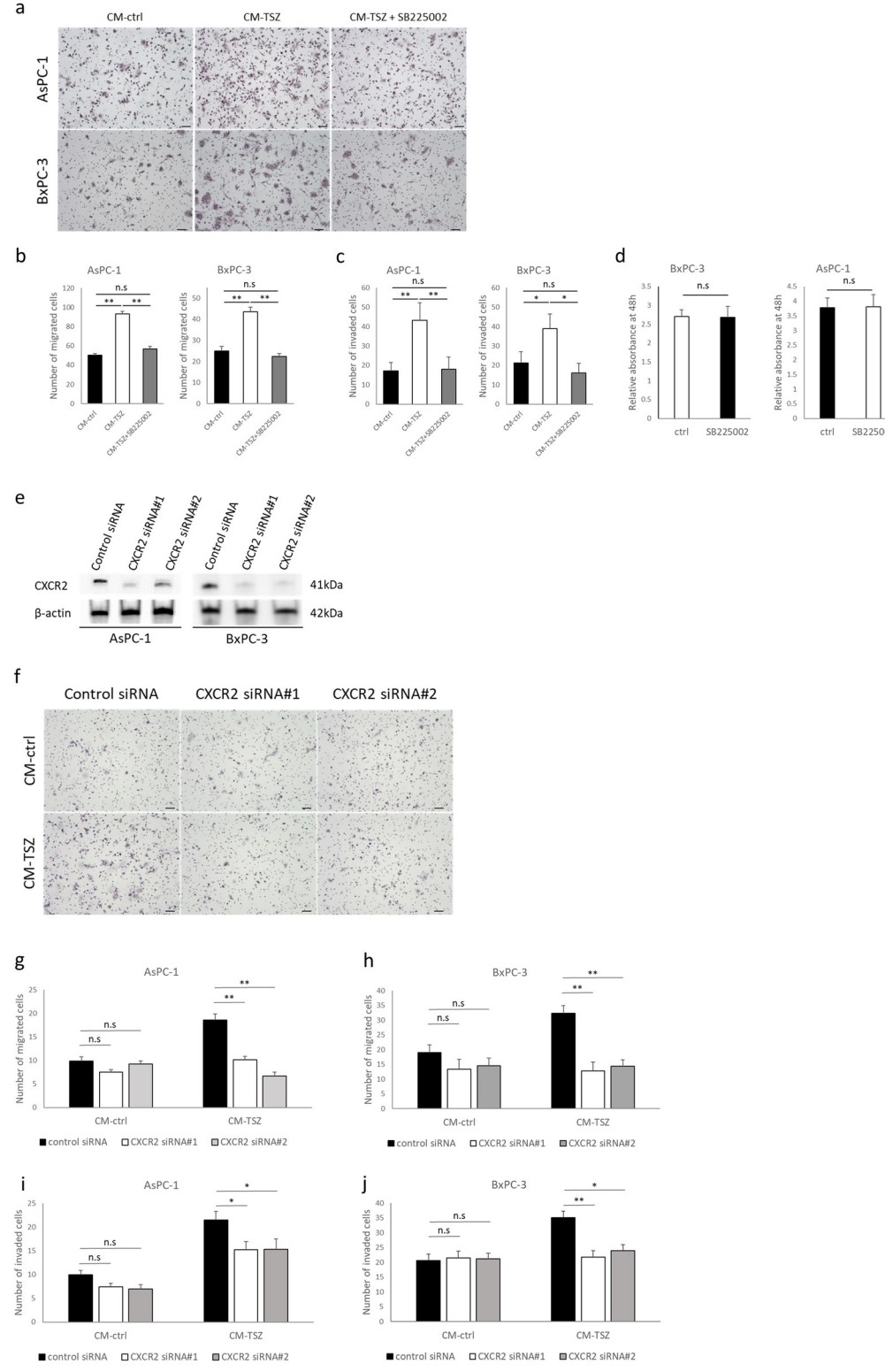

**Fig 5. Effect of CXCR2 inhibition by selective antagonist, SB225002, or knockdown with siRNA in PC cells.** (a-d) The inhibitory effect of SB225002 (10 nM) on CXCR2 in pancreatic cancer cells was enhanced by conditioned medium from necroptotic cells, and is shown through Transwell migration assay, Matrigel invasion assay and proliferation assay. (a) Representative images of Transwell migration assay. (b) Quantitative data of migrated cells. (c) Quantitative data of invaded cells in Matrigel invasion assay. (d) Effect of SB225002 on pancreatic cancer cell proliferation after 48

hours. Absorbance relative to 0 hour. (e-j) CXCR2 knockdown with siRNA in PC cells. (e) CXCR2 silencing was confirmed by western blot analysis. (f) Representative images of Transwell migration assay. Quantitative data of Transwell migration assays performed with (g) AsPC-1 and (h) BxPC-3 cells, and Matrigel invasion assays performed with (i) AsPC-1 and (j) BxPC-3 cells downregulated for CXCR2 with siRNA. Graph show mean ± SE. *P<0.05; *P<0.01; n.s: not significant.

BxPC-3 cells (Fig 7D). The response of CXCR2 expression to TSZ treatment was the same as above (S1E Fig). The response of CXCR2 expression to necroptotic stimulants appeared to be different based on the cell type. It may be related to the responsiveness to drugs or the degree of appearance of side effects.

## Discussion

The key effector molecules of necroptosis are reportedly downregulated in various types of cancer. For example, RIP3 expression is downregulated in human acute myeloid leukemia cells, RIP1 and RIP3 are significantly decreased in colon cancer tissue, and DNA methylation reduced RIP3 expression in breast and lung cancer cells [38–41]. Low MLKL expression in early-stage resected PC has also been associated with poor prognosis [42]. However, Liu et al [43]. reported that expression of phosphorylated MLKL was correlated with lower survival rates in esophageal and colon cancers. Seifert et al [44]. reported that RIP1 and RIP3 were highly expressed in human PC, and RIP3 depletion suppressed tumorigenesis. Strilic et al [45]. showed that endothelial cell necroptosis promotes cancer cell extravasation and metastasis.

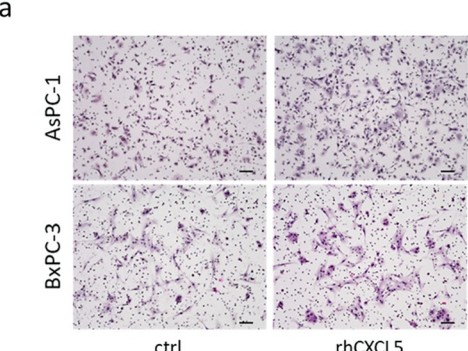

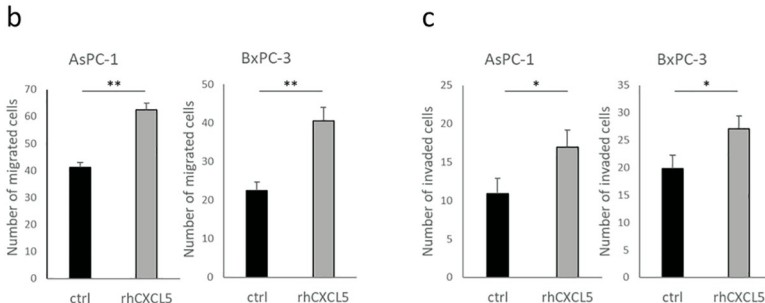

**Fig 6. Effect of human recombinant CXCL5 (rh CXCL5) on PC cell motility.** PC cell motility was evaluated by Transwell migration and Transwell-Matrigel invasion assays. PC cells pretreated with rh CXCL5 (10 ng/ml) or vehicle for 12 hours. (a) Representative images of migrated cells. scale bars = 100 μm. (b) Quantitative data of migrated cells. (c) Quantitative data of invaded cells. Graphs show mean ± SE. *P < 0.05; **P<0.01.

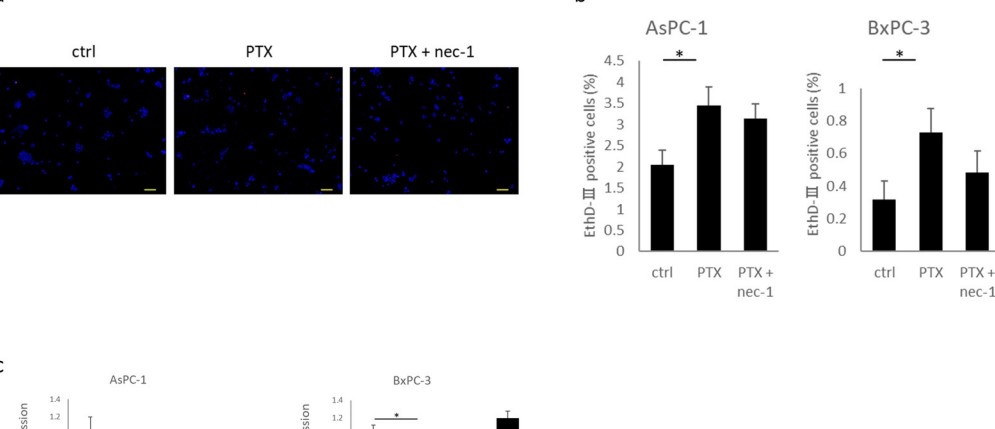

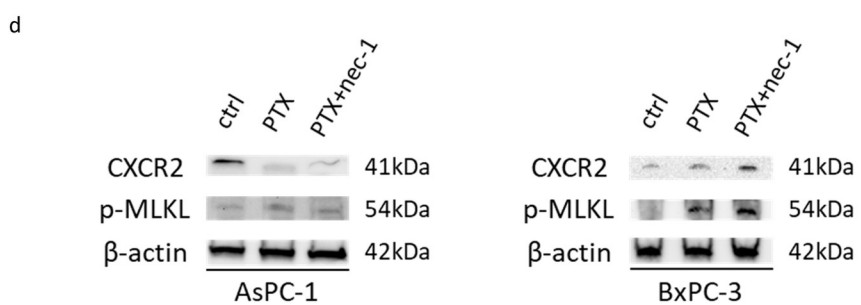

**Fig 7. Induction of cell death in human PC cells by chemotherapeutic agents, and detection of CXCR2.** (a) Fluorescence images of pancreatic cancer cells treated with PTX and PTX + nec-1. Blue; Hoechst 33342. Red; EthD-III. Scale bars = 100 μm. (b) Rate of dead cells treated with each agent for 12 hours. (c) Analysis of CXCR2 mRNA expression in PC cells treated with programmed cell death-inductive agents. Results are shown as relative to gene expression in ctrl after normalization against 18S rRNA. Cells were treated with TSZ, PTX (40 nM), or PTX (40 nM) + nec-1 (30 μM) for 12 hours. (d) Western blot analyses of p-MLKL and CXCR2. Graph shows the mean ± SE. *$P<0.01$.

Thus, the role of necroptosis in cancer is controversial. In this study, we showed that the necrosome components, RIP3 and MLKL, are highly expressed in human PC. Particularly, the main executor of necroptosis, MLKL, was highly expressed at the invasion front of the tumor. In the *in vitro* experiment, we induced necroptosis only in PC cells with high MLKL expression. Therefore, high expression of MLKL at the tumor invasion front may induce necroptosis.

When we induced necroptosis in PC cells, TNF-α was used as a trigger. The PC microenvironment provides some sources of TNF-α, such as macrophages, adipocytes, and fibroblasts [46]. These cells may trigger necroptosis in PC with high MLKL expression. Furthermore, CM of necroptotic cells promoted PC cell migration and invasion. We found that CXCL5 expression was upregulated by necroptotic cell-derived CM, and expression of its receptor, CXCR2, was upregulated in PC cells compared with non-cancerous HPDE cells. CXCR2 is a member of the G-protein-coupled chemokine receptor family. The C-X-C-motif chemokine CXCL5 and IL-8 bind to CXCR2 specifically. Recent studies suggest that CXCR2 plays a crucial role in

invasion, angiogenesis, and metastases of various cancer types such as prostate, lung, colon, oral, and pancreatic cancers [47–51]. Steele et al [36]. revealed CXCR2 expression at the PC tumor border, and that high CXCR2 expression was associated with poor outcomes. In our study, CXCR2 was also expressed strongly at the invasive front of the tumor. Inhibition of CXCR2 using SB225002 impeded cancer cell migration and invasion enhanced by necroptosis. Hence, CXCR2 has a critical role in the mechanism of necroptosis-enhanced tumor migration and invasion. These findings suggest that necroptosis promotes cancer progression locally via CXCL5 and CXCR2. However, most investigations indicate a defensive role of necroptosis against cancer [46]. Possibly, necroptosis switches functions in cancer depending on the frequency or localization. Therefore, its role in PC warrants further study.

Evasion of apoptosis is a mechanism of resistance to conventional cancer therapies, but necroptosis involves the fail-safe mechanism of apoptosis [52]. Preclinical studies reported that necroptosis was induced even in apoptosis-resistant tumors [53,54]. Induction of necroptosis has become as an attractive therapeutic strategy against cancer. Several therapeutic agents were shown to kill cancer cells via necroptosis, including Taxol and 5-FU [55,56]. Induction of necroptosis may become a critical strategy against apoptosis-resistant tumors, but its progressive role in cancer should be considered. Our present results suggest that CXCR2 is a potential therapeutic target, combined with treatment to induce necroptosis.

In conclusion, we have shown that key mediators of necroptosis were expressed in PC, and CXCL5 released from necroptotic cells promotes cancer cell migration and invasion via CXCR2 in PC cells. Induction of necroptosis is a possible therapeutic strategy to overcome apoptosis resistance in PC, but it could also enhance cancer progression. Development of necroptosis induction therapy might include a means of suppressing its negative influences.

## Supporting information

**S1 Fig. Induction of necroptosis in AsPC-1 cells by FLZ treatment, and detection of CXCL5 in CM-FLZ.** FLZ: Fas ligand, LCL161 (Smac mimetic), zIETD-FMK. (a) Fluorescent images of AsPC-1 cells treated with combination of recombinant human FAS ligand, LCL161, zIETD-FMK and nec-1. EthD-III staining shows dead cells. Scale bars = 100 μm. (b) Frequency of dead cells treated with each agent for 12 hours. (c) Western blot analysis of p-MLKL, MLKL and cleaved caspase-8. AsPC-1 cells treated with FLZ for 4 hours. (d) Concentration of CXCL5 in CM-ctrl, CM-FLZ and CM-FLZ+nec-1 measured by ELISA. (e) Western blot analysis of CXCL5 and CXCR2 expression in TSZ-treated AsPC-1 and BxPC-3 cells over time. Graphs show mean ± SE. $^*P < 0.05$; $^{**}P < 0.01$.
(TIF)

**S2 Fig. Detection of chemokines and chemokine receptors in human PC tissues and cell lines.** (a) Representative images of CXCR2 expression at the invasive front and center of the tumor (scale bars = 100 μm). (b) Comparison of the CXCR2 staining intensity in pancreatic cancer cells at the invasive front and center of the tumor. Five fields of view at ×200 magnification per patient were analyzed in 21 patients. (c) Representative images of CXCL5 immunohistochemistry in human PC tissues. (d) Representative image of CXCL1 immunohistochemistry in human PC. (e, f) Comparison of the CXCR2 and MLKL staining intensity at the tumor invasive front in patients who received (n = 5) or did not receive (n = 16) preoperative chemotherapy. Five fields of view at ×200 magnification per patient were analyzed. $^*P < 0.01$.
(TIF)

**S1 Raw images.**
(PDF)

## Acknowledgments

The authors thank E. Manabe, S. Sadatomi, N Torada (Department of Surgery and Oncology, Kyushu University Hospital), and members of the Research Support Center and Department of Anatomic Pathology, Graduate School of Medical Sciences, Kyushu University for their expert technical assistance, and A. Doi (Cell Innovator Co., Ltd., Fukuoka, Japan) for assistance with gene expression analysis. We also thank Marla Brunker, from Edanz Group (www.edanzediting.com/ac) for editing a draft of this manuscript.

## Author Contributions

**Conceptualization:** Yohei Ando, Kenoki Ohuchida.

**Data curation:** Yohei Ando, Shin Takesue.

**Formal analysis:** Yohei Ando.

**Funding acquisition:** Kenoki Ohuchida.

**Investigation:** Yohei Ando.

**Methodology:** Yohei Ando, Shin Kibe, Toshiya Abe, Chika Iwamoto.

**Project administration:** Yohei Ando, Kenoki Ohuchida, Koji Shindo, Taiki Moriyama, Kohei Nakata.

**Resources:** Kenoki Ohuchida, Yoshiki Otsubo, Yoshinao Oda.

**Supervision:** Takao Ohtsuka, Masafumi Nakamura.

**Validation:** Yoshihiro Miyasaka.

**Writing – original draft:** Yohei Ando.

**Writing – review & editing:** Kenoki Ohuchida.

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
