## [Decision Letter · Decision Letter 0]

11 Nov 2019

PONE-D-19-24224

Necroptosis in pancreatic cancer promotes cancer cell migration and invasion by release of CXCL5

PLOS ONE

Dear Dr. Ohuchida,

Thank you for submitting your manuscript to PLOS ONE. After careful consideration, we feel that it has merit but does not fully meet PLOS ONE’s publication criteria as it currently stands. Therefore, we invite you to submit a revised version of the manuscript that addresses the points raised during the review process.

We would appreciate receiving your revised manuscript by Dec 26 2019 11:59PM. To enhance the reproducibility of your results, we recommend that if applicable you deposit your laboratory protocols in protocols.io, where a protocol can be assigned its own identifier (DOI) such that it can be cited independently in the future. For instructions see: http://journals.plos.org/plosone/s/submission-guidelines#loc-laboratory-protocols

We look forward to receiving your revised manuscript.

Kind regards,

Yi-Hsien Hsieh, Ph.D.

Academic Editor

PLOS ONE

Journal Requirements:

4. Thank you for stating the following above the Abstract of your manuscript:

'Grant Support

This work was supported in part by Japan Society for the Promotion of Science Grants

in-Aid for Scientific Research (Grant Numbers 17H04284, 17K19602, 18H02881,

18K08708, 16H05418, 17K19605).'

'The authors received no specific funding for this work.'

Please provide an amended Funding Statement that declares *all* the funding or sources of support received during this specific study (whether external or internal to your organization) as detailed online in our guide for authors at http://journals.plos.org/plosone/s/submit-nowPlease state what role the funders took in the study.  If any authors received a salary from any of your funders, please state which authors and which funder. If the funders had no role, please state: "The funders had no role in study design, data collection and analysis, decision to publish, or preparation of the manuscript."

Reviewers' comments:

Reviewer's Responses to Questions

**Comments to the Author**

1. Is the manuscript technically sound, and do the data support the conclusions?

Reviewer #1: Partly

Reviewer #2: Yes

2. Has the statistical analysis been performed appropriately and rigorously? 

Reviewer #1: Yes

Reviewer #2: Yes

3. Have the authors made all data underlying the findings in their manuscript fully available?

Reviewer #1: Yes

Reviewer #2: Yes

4. Is the manuscript presented in an intelligible fashion and written in standard English?

Reviewer #1: Yes

Reviewer #2: Yes

5. Review Comments to the Author

Reviewer #1: Pancreatic cancer is an aggressive disease often discovered at a late stage, and treatment options are limited. Better understanding of the determinants of PC progression may lead to more effective treatments. This research article suggests possible mechanisms by which necroptosis in PC cells may contribute to tumor invasiveness, with a focus on impacts to tumor epithelium itself and not on tumor-associated macrophages or neutrophils, development of immune responses, or angiogenesis. The topic is of general interest, and the writing is clear. Note that there are many published observations regarding CXCL5, CXCR2, and CXCL1 in PC, and the authors should be meticulous in positioning their work in the context of what has already been shown.

1) The authors begin with the observation of RIP3 and MLKL expression in human PC tissue (previously published by Seifert et al 2016 Nature) then extend the observation by stating the detection is at the leading, invasive edge of the tumor. Steele et al (Cancer Cell 2016) previously showed CXCR2 expression at the PC tumor border. What was the expression pattern for CXCR2, CXCL5, and CXCL1 in the PC tissue panel? Which patients received chemotherapy, and was there any relationship to the expression of RIP3, MLKL, or the chemokines and receptor of interest?

2) Li et al (AJ Pathol 2011 178:1342) and Matsuo (Int J Cancer 2009 125: 1032) show differences between PC cells in their baseline expression of CXCL5. Matsuo et al additionally show substantially stronger expression of CXCL1 compared with CXCL5 in untreated PC lines. Does the CXCL5 released into conditioned medium after death-inducing stimulus represent stored chemokine that is released when membrane integrity is lost? Or is CXCL5 expression induced in cells undergoing programmed cell death, in the bystanders not undergoing cell death, or both? A repeat of immunofluorescence staining as performed in Figure 2, with a time course that includes CXCL5 detection, would clarify this point. Is CXCL1 also released into conditioned medium? Is expression of CXCL2 impacted by necroptotic stimuli? Is the in vitro activity of conditioned medium dependent on CXCL1?

3) The in vitro analysis would be strengthened with the inclusion of a relevant, necroptosis-inducing chemotherapeutic, for comparison with the experimental TSZ necroptosis induction strategy chosen. Any changes in CXCR2 expression related to treatment in the chosen cell lines should be shown (immunoblot, mRNA).

4) The number of times in vitro experiments were performed is missing in places.

Reviewer #2: The authors examined roles of necroptosis of human pancreatic cancer cells by immunohistochemistry of cancer samples and in vitro experiments. They show that CXCL5 is a key mediator from necroptotic pancreatic cancer cells and promotes invasion and migration of cancer cells via the CXCR5/CXCR2 axis. This result suggests that, although necroptosis of cancer cells is an anti-cancer effect, this type of cell death could enhance cancer progression. The reviewer supposes that this information is useful for the readers of PLOS ONE.

Minor comments:

1) Page 22, line 354 and 355. Two different description, such as S1 Figs a-c and Supplementary Fig 1d.

2) Page 19, line 297-298. Cleaved caspase-8 was upregulated in TS-treated PANC-1 cells (Fig. 2d). However, it seems no change.

6. PLOS authors have the option to publish the peer review history of their article (what does this mean?). If published, this will include your full peer review and any attached files.

Reviewer #1: No

Reviewer #2: No

---

## [Author Response · Author response to Decision Letter 0]

26 Dec 2019

Responses to Reviewer #1

Thank you for your review of our manuscript. We have responded to each of your points below.

1) The authors begin with the observation of RIP3 and MLKL expression in human PC tissue (previously published by Seifert et al 2016 Nature) then extend the observation by stating the detection is at the leading, invasive edge of the tumor. Steele et al (Cancer Cell 2016) previously showed CXCR2 expression at the PC tumor border. What was the expression pattern for CXCR2, CXCL5, and CXCL1 in the PC tissue panel? Which patients received chemotherapy, and was there any relationship to the expression of RIP3, MLKL, or the chemokines and receptor of interest?

Response: We observed the expression pattern of CXCR2 in PC. Consistent with previous reports, CXCR2 was strongly expressed at the invasive front of tumors. CXCL5 expression was varied among samples, but we did not observe characteristic changes due to localization or neoadjuvant chemotherapy. CXCL1 was mainly expressed in stromal cells adjacent to the tumor than in tumor cells. Although stromal cells expressing CXCL1 may have some effect on tumors via CXCR2, we focused on tumor cells in this study, which needs further study.

To clarify the influence of chemotherapy on the PC cell expression pattern of MLKL, RIP3, chemokines, and receptors, we performed additional immunohistochemistry in patients who received neoadjuvant chemotherapy. CXCR2 expression at the invasive front was slightly reduced in patients treated with neoadjuvant chemotherapy. However, we did not observe significant changes in the expression patterns, localization, or staining intensity of other proteins.

We have modified the manuscript as follows.

Page 23, line 368

Next, we observed expression patterns of CXCR2 and CXCL5 in human PC tissues. Consistent with a previous report [36], CXCR2 expression was evident at the invasive front of the tumor (S2 Figs a and b). Although CXCL5 expression varied among patients (S2 Fig c), we did not observe significant changes due to localization or preoperative chemotherapy. CXCL1 is a secreted growth factor that signals through CXCR2. We assessed CXCL1 expression in PC tissues. Interestingly, CXCL1 was mainly expressed in stromal cells around the tumor than in tumor cells (S2 Fig d). Stromal cells expressing CXCL1 may have some effect on tumors via CXCR2. To clarify the influence of chemotherapy on the PC cell expression pattern of CXCR2 and MLKL we compared patients who received or did not receive neoadjuvant chemotherapy (NAC). CXCR2 expression at the invasive front was slightly reduced in patients treated with neoadjuvant chemotherapy, but changes of MLKL expression was not significant (S2 Figs e and f).

Page 28, line 475

We found that CXCL5 expression was upregulated by necroptotic cell-derived CM, and expression of its receptor, CXCR2, was upregulated in PC cells compared with non-cancerous HPDE cells. CXCR2 is a member of the G-protein-coupled chemokine receptor family. The C-X-C-motif chemokine CXCL5 and IL-8 bind to CXCR2 specifically. Recent studies suggest that CXCR2 plays a crucial role in invasion, angiogenesis, and metastases of various cancer types such as prostate, lung, colon, oral, and pancreatic cancers [47–51]. Steele et al [36]. revealed CXCR2 expression at the PC tumor border, and that high CXCR2 expression was associated with poor outcomes. In our study, CXCR2 was also expressed strongly at the invasive front of the tumor. Inhibition of CXCR2 using SB225002 impeded cancer cell migration and invasion enhanced by necroptosis. Hence, CXCR2 has a critical role in the mechanism of necroptosis-enhanced tumor migration and invasion. These findings suggest that necroptosis promotes cancer progression locally via CXCL5 and CXCR2. However, most investigations indicate a defensive role of necroptosis against cancer [46]. Possibly, necroptosis switches functions in cancer depending on the frequency or localization. Therefore, its role in PC warrants further study.

Page 43, line 713 

S2 Fig. Detection of chemokines and chemokine receptors in human PC tissues and cell lines. 

(a) Representative images of CXCR2 expression at the invasive front and center of the tumor (scale bars = 100 μm). (b) Comparison of the CXCR2 staining intensity in pancreatic cancer cells at the invasive front and center of the tumor. Five fields of view at ×200 magnification per patient were analyzed in 21 patients. (c) Representative images of CXCL5 immunohistochemistry in human PC tissues. (d) Representative image of CXCL1 immunohistochemistry in human PC. (e, f) Comparison of the CXCR2 and MLKL staining intensity at the tumor invasive front in patients who received (n=5) or did not receive (n=16) preoperative chemotherapy. Five fields of view at ×200 magnification per patient were analyzed. *P<0.01.

2) Li et al (AJ Pathol 2011 178:1342) and Matsuo (Int J Cancer 2009 125: 1032) show differences between PC cells in their baseline expression of CXCL5. Matsuo et al additionally show substantially stronger expression of CXCL1 compared with CXCL5 in untreated PC lines. Does the CXCL5 released into conditioned medium after death-inducing stimulus represent stored chemokine that is released when membrane integrity is lost? Or is CXCL5 expression induced in cells undergoing programmed cell death, in the bystanders not undergoing cell death, or both? A repeat of immunofluorescence staining as performed in Figure 2, with a time course that includes CXCL5 detection, would clarify this point. Is CXCL1 also released into conditioned medium? Is expression of CXCL2 impacted by necroptotic stimuli? Is the in vitro activity of conditioned medium dependent on CXCL1?

Response: We have attempted to perform immunofluorescence staining of CXCL5 over a time course, but it has not been successful so far. Instead, we detected temporal changes of CXCL5 levels in PC cells treated with TSZ by western blot analysis.

CXCL5 expression in PC cells was increased after TSZ treatment. It suggest that CXCL5 was induced in the process of programmed cell death and released into the conditioned medium. Unfortunately, it is unclear whether cells undergoing programmed cell death or bystanders release this chemokine. CXCL1 in conditioned medium was detected by protein array analysis (Fig 4a), but there were no significant changes after TSZ treatment. Therefore, CXCL1 may not influence the activity of conditioned medium derived from necroptotic cells. Furthermore, CXCL2 expression was not changed by TSZ treatment.

We have modified the manuscript text as follows.

Page 22, line 362 

To clarify whether the CXCL5 released into the conditioned medium after death-inducing stimulus was stored in cells or induced in cells undergoing programmed cell death, we treated PC cells with TSZ and detected temporal changes of CXCL5 levels in PC cells. CXCL5 expression in PC cells was increased after TSZ treatment (S1 Fig e). These results suggest that CXCL5 was induced in the process of programmed cell death and released into the conditioned medium. 

3) The in vitro analysis would be strengthened with the inclusion of a relevant, necroptosis-inducing chemotherapeutic, for comparison with the experimental TSZ necroptosis induction strategy chosen. Any changes in CXCR2 expression related to treatment in the chosen cell lines should be shown (immunoblot, mRNA). 

Response: paclitaxel (PTX) has been reported to induce necroptosis in cancer cells. We treated PC cells with PTX or PTX + nec-1. PTX treatments induced cell death and p-MLKL expression in PC cells, but inhibition by nec-1 was not significant. PTX treatment may induces some necroptosis in PC, but that’s not main effect. We did not detect significant changes in CXCR2 mRNA expression levels after these treatments. However, in western blot analysis, CXCR2 expression was reduced in AsPC-1 cells by these treatments, but increased in BxPC-3 cells. The response of CXCR2 expression to the TSZ treatment was the same as above. The response of CXCR2 expression to necroptotic stimulants appeared to be different based on the cell type. It may be related to the responsiveness to drugs or the degree of appearance of side effects.

We have modified the manuscript text as follows.

Page 10, line 141

To evaluate the effects of chemotherapeutic agents on chemokine receptor expression in PC cells, we treated PC cells by paclitaxel (FUJIFILM Wako Chemicals, Osaka, Japan; 40nM) or paclitaxel (40nM) + necrostatin-1 (30μM) for 12hours.

Page 26, line 431

Necroptosis-inducing chemotherapeutic agent changes CXCR2 expression in PC cells.

Paclitaxel (PTX) has been reported to induce necroptosis in cancer (37). We treated PC cells with PTX or PTX + nec-1 for 12hours and then detected changes in CXCR2 expression. PTX treatments induced cell death and p-MLKL expression in AsPC-1 and BxPC-3 cells, but inhibition by nec-1 was not significant (Fig 7s a, b and d). We did not detect significant changes in CXCR2 mRNA expression levels after these treatments (Fig 7 c). However, in western blot analysis CXCR2 expression was reduced in AsPC-1 cells by these treatments, but increased in BxPC-3 cells (Fig 7 d). The response of CXCR2 expression to TSZ treatment was the same as above (S1 Fig e). The response of CXCR2 expression to necroptotic stimulants appeared to be different based on the cell type. It may be related to the responsiveness to drugs or the degree of appearance of side effects.

Page 27, line 444

Fig. 7. Induction of cell death in human PC cells by chemotherapeutic agents and detection of CXCR2.

(a) Fluorescence images of pancreatic cancer cells treated with PTX and PTX + nec-1. Blue; Hoechst 33342. Red; EthD-Ⅲ. Scale bars = 100 μm. (b) Rate of dead cells treated with each agent for 12 hours. (c) Analysis of CXCR2 mRNA expression in PC cells treated with programmed cell death-inductive agents. Results are shown as relative to gene expression in ctrl after normalization against 18S rRNA. Cells were treated with TSZ, PTX (40 nM), or PTX (40 nM) + nec-1 (30 μM) for 12 hours. (d) Western blot analyses of p-MLKL and CXCR2. Graph shows the mean ± SE. *P<0.01.

4) The number of times in vitro experiments were performed is missing in places.

Response: We have added the number of times in vitro experiments were performed in the Materials and methods.

Response to Reviewer #2

Thank you for your comments. Our responses to your comments are as follows. 

Minor comments:

1) Page 22, line 354 and 355. Two different description, such as S1 Figs a-c and Supplementary Fig 1d.

We have modified the manuscript text as follows.

Page 22, line 359

CXCL5 upregulation in CM-FLZ from AsPC-1 cells was detected by an ELISA (S1 Fig d).

2) Page 19, line 297-298. Cleaved caspase-8 was upregulated in TS-treated PANC-1 cells (Fig. 2d). However, it seems no change.

Response: We detected other activated fragment of cleaved caspase-8, p43/p41, which appears first after activation. Cleaved caspase-8 p43/41 shows the difference more clearly.

---

## [Decision Letter · Decision Letter 1]

7 Jan 2020

Necroptosis in pancreatic cancer promotes cancer cell migration and invasion by release of CXCL5

PONE-D-19-24224R1

Dear Dr. Ohuchida,

We are pleased to inform you that your manuscript has been judged scientifically suitable for publication and will be formally accepted for publication once it complies with all outstanding technical requirements.

With kind regards,

Yi-Hsien Hsieh, Ph.D.

Academic Editor

PLOS ONE

Additional Editor Comments (optional):

Reviewers' comments:

Reviewer's Responses to Questions

**Comments to the Author**

1. If the authors have adequately addressed your comments raised in a previous round of review and you feel that this manuscript is now acceptable for publication, you may indicate that here to bypass the “Comments to the Author” section, enter your conflict of interest statement in the “Confidential to Editor” section, and submit your "Accept" recommendation.

Reviewer #1: All comments have been addressed

Reviewer #2: All comments have been addressed

2. Is the manuscript technically sound, and do the data support the conclusions?

Reviewer #1: Yes

Reviewer #2: Yes

3. Has the statistical analysis been performed appropriately and rigorously? 

Reviewer #1: Yes

Reviewer #2: Yes

4. Have the authors made all data underlying the findings in their manuscript fully available?

Reviewer #1: Yes

Reviewer #2: Yes

5. Is the manuscript presented in an intelligible fashion and written in standard English?

Reviewer #1: Yes

Reviewer #2: Yes

6. Review Comments to the Author

Reviewer #1: The authors have made a good faith effort to address the comments provided to them. Additional references cited for context and considerable additional data have been incorporated into the manuscript. The manuscript provides a more complete picture of CXCL5, CCR2, and CXCL1 expression in tumor tissues and in PC lines challenged with necroptosis-inducing stimuli. Additional clarity has been provided in figure legends. The current version of the manuscript meets publication criteria.

Reviewer #2: MS# PLOS-D-19-24224R1

The authors respond to my comments properly. The reviewer supposes that this study is worth publishing in the PLOS ONE.

7. PLOS authors have the option to publish the peer review history of their article (what does this mean?). If published, this will include your full peer review and any attached files.

Reviewer #1: No

Reviewer #2: No

---

## [Editor Report · Acceptance letter]

10 Jan 2020

PONE-D-19-24224R1 

Necroptosis in pancreatic cancer promotes cancer cell migration and invasion by release of CXCL5 

Dear Dr. Ohuchida:

I am pleased to inform you that your manuscript has been deemed suitable for publication in PLOS ONE. Congratulations! Your manuscript is now with our production department. 

With kind regards,

on behalf of

Dr Yi-Hsien Hsieh 

Academic Editor

PLOS ONE